# PV3D: A 3D Generative Model for Portrait Video Generation

**Eric Zhongcong Xu**[1*], **Jianfeng Zhang**[2*], **Jun Hao Liew**[2], **Wenqing Zhang**[2],
**Song Bai**[2], **Jiashi Feng**[2], **Mike Zheng Shou**[1†]
[1] Show Lab, National University of Singapore   [2] ByteDance
{zhongcongxu,zhangjianfeng}@u.nus.edu
{junhao.liew,wenqingzhang,jshfeng}@bytedance.com
{songbai.site, mike.zheng.shou}@gmail.com

## ABSTRACT

Recent advances in generative adversarial networks (GANs) have demonstrated the capabilities of generating stunning photo-realistic portrait images. While some prior works have applied such image GANs to unconditional 2D portrait video generation and static 3D portrait synthesis, there are few works successfully extending GANs for generating 3D-aware portrait videos. In this work, we propose PV3D, the first generative framework that can synthesize multi-view consistent portrait videos. Specifically, our method extends the recent static 3D-aware image GAN to the video domain by generalizing the 3D implicit neural representation to model the spatio-temporal space. To introduce motion dynamics into the generation process, we develop a motion generator by stacking multiple motion layers to synthesize motion features via modulated convolution. To alleviate motion ambiguities caused by camera/human motions, we propose a simple yet effective camera condition strategy for PV3D, enabling both temporal and multi-view consistent video generation. Moreover, PV3D introduces two discriminators for regularizing the spatial and temporal domains to ensure the plausibility of the generated portrait videos. These elaborated designs enable PV3D to generate 3D-aware motion-plausible portrait videos with high-quality appearance and geometry, significantly outperforming prior works. As a result, PV3D is able to support downstream applications such as static portrait animation and view-consistent motion editing. Code and models are available at https://showlab.github.io/pv3d.

## 1   INTRODUCTION

Recent progress in generative adversarial networks (GANs) has led human portrait generation to unprecedented success (Karras et al., 2020; 2021; Skorokhodov et al., 2022) and has spawned a lot of industrial applications (Tov et al., 2021; Richardson et al., 2021). Generating portrait videos has emerged as the next challenge for deep generative models with wider applications like video manipulation (Abdal et al., 2022) and animation (Siarohin et al., 2019). A long line of work has been proposed to either learn a direct mapping from latent code to portrait video (Vondrick et al., 2016; Saito et al., 2017) or decompose portrait video generation into two stages, *i.e.*, content synthesis and motion generation (Tian et al., 2021; Tulyakov et al., 2018; Skorokhodov et al., 2022).

Despite offering plausible results, such methods only produce 2D videos without considering the underlying 3D geometry, which is the most desirable attribute with broad applications such as portrait reenactment (Doukas et al., 2021), talking face animation (Siarohin et al., 2019), and VR/AR (Cao et al., 2022). Current methods typically create 3D portrait videos through classical graphics techniques (Wang et al., 2021b; Ma et al., 2021; Grassal et al., 2022), which require multi-camera systems, well-controlled studios, and heavy artist works. In this work, we aim to alleviate the effort of creating high-quality 3D-aware portrait videos by learning from 2D monocular videos only, **without** the need of any 3D or multi-view annotations.

---

*Equal contribution, work done during an internship at ByteDance
†Corresponding author

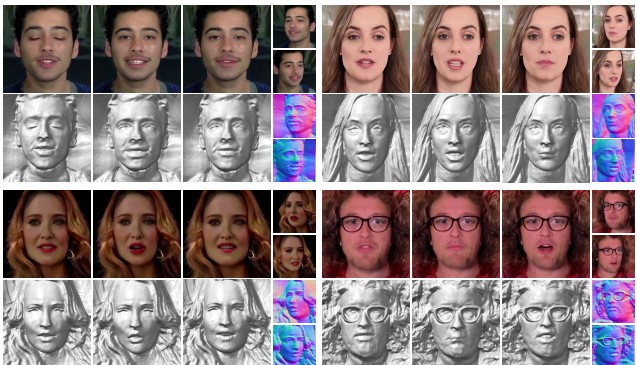

Figure 1: Our PV3D can generate photo-realistic portrait videos with diverse motions and dynamic 3D geometry. We render surfaces extracted by marching cubes. The video frames and shape (normal map) can be rendered from arbitrary viewpoints. Please see our project page for video results.

Recent 3D-aware portrait generative methods have witnessed rapid advances (Schwarz et al., 2020; Gu et al., 2022; Chan et al., 2021; Niemeyer & Geiger, 2021; Or-El et al., 2022; Chan et al., 2022). Through integrating implicit neural representations (INRs) (Sitzmann et al., 2020; Mildenhall et al., 2020) into GANs (Karras et al., 2019; 2020), they can produce photo-realistic and multi-view consistent results. However, such methods are limited to static portrait generation and can hardly be extended to portrait video generation due to several challenges: 1) it remains unclear how to effectively model 3D dynamic human portrait in a generative framework; 2) learning dynamic 3D geometry without 3D supervision is highly under-constrained; 3) entanglement between camera movements and human motions/expressions introduces ambiguities to the training process. To this end, we propose a **3D P**ortrait **V**ideo generation model (PV3D), the first method that can generate high-quality 3D portrait videos with diverse motions while learning purely from monocular 2D videos. PV3D enables 3D portrait video modeling by extending 3D tri-plane representation (Chan et al., 2022) to the spatio-temporal domain. In this paper, we comprehensively analyze various design choices and arrive at a set of novel designs, including decomposing latent codes into appearance and motion components, temporal tri-plane based motion generator, proper camera pose sequence conditioning, and camera-conditioned video discriminators, which can significantly improve the video fidelity and geometry quality for 3D portrait video generation.

As shown in Figure 1, despite being trained from only monocular 2D videos, PV3D can generate a large variety of photo-realistic portrait videos under arbitrary viewpoints with diverse motions and high-quality 3D geometry. Comprehensive experiments on various datasets including Vox-Celeb (Nagrani et al., 2017), CelebV-HQ (Zhu et al., 2022) and TalkingHead-1KH (Wang et al., 2021a) well demonstrate the superiority of PV3D over previous state-of-the-art methods, both qualitatively and quantitatively. Notably, it achieves 29.1 FVD on VoxCeleb, improving upon a concurrent work 3DVidGen (Bahmani et al., 2022) by 55.6%. PV3D can also generate high-quality 3D geometry, achieving the best multi-view identity similarity and warping error across all datasets.

Our contributions are three-fold. 1) To our best knowledge, PV3D is the first method that is capable to generate a large variety of 3D-aware portrait videos with high-quality appearance, motions, and geometry. 2) We propose a novel temporal tri-plane based video generation framework that can synthesize 3D-aware portrait videos by learning from 2D videos only. 3) We demonstrate state-of-the-art 3D-aware portrait video generation on three datasets. Moreover, our PV3D supports several downstream applications, *i.e.*, static image animation, monocular video reconstruction, and multi-view consistent motion editing.

## 2    RELATED WORK

**2D video generation.** Early video generation works (Vondrick et al., 2016; Saito et al., 2017) propose to learn a video generator to transform random vectors directly to video clips. While recent video generation works adopt a similar paradigm to design the video generator, *i.e.*, disentangle the video content and motion (trajectory), then control them by different random noises. For the video

content, most of the works build their frameworks on top of generative adversarial networks (GAN) designed for image domain, such as StyleGAN (Karras et al., 2019) and INR-GAN (Skorokhodov et al., 2021). Based on image GANs, video GAN works further extend the generation process to temporal domain using various motion generation approaches. MoCoGAN (Tulyakov et al., 2018) and its following work MoCoGAN-HD (Tian et al., 2021) generate the motion code sequence autoregressively, which is implemented as random process. StyleGAN-V (Skorokhodov et al., 2022) also generates a motion code sequence for each frame, while the motion codes are sampled separately and thus its generator can synthesize video frames independently. In contrast, DiGAN (Yu et al., 2022) only sample one motion code for the entire video and the motion for each frame is generated by the INR network using time instant. This compact video code design is in line with the property of temporal consistency, *e.g.*, a talking person only move the lips or twinkles, while the face shape does not change rapidly (Tewari et al., 2019).

**3D-aware generation.** Image GANs (Karras et al., 2020; 2021) have demonstrated impressive capability in synthesizing high-resolution photo-realistic images. By incorporating implicit neural representations (Mildenhall et al., 2020; Sitzmann et al., 2020) or differentiable neural rendering (Kato et al., 2018) into GANs, recent works (Schwarz et al., 2020; Niemeyer & Geiger, 2021; Chan et al., 2021; Shi et al., 2021) can produce multi-view consistent images. However, most of them are limited at image quality and resolution due to the heavy computation cost of traditional 3D representations. Follow-up works are proposed to address this issue by either developing an efficient 3D representation (Chan et al., 2022; Deng et al., 2022; Schwarz et al., 2022; Zhao et al., 2022) or dividing image generation into two stages (Gu et al., 2022; Or-El et al., 2022; Zhang et al., 2022b), *i.e.*, generating low-resolution images by volume rendering (Max, 1995) and then refine them using super-resolution approaches. However, these methods are limited to static object generation. Recently, CoRF (Zhuang et al., 2022) proposes to condition the static 3D GAN on estimated motion features. In addition, a concurrent work 3DVidGen (Bahmani et al., 2022) has been proposed to extend unconditional 3D-aware generation into video domain. Although 3DVidGen can generate plausible results with changing viewpoints, its video fidelity and multi-view consistency are unsatisfactory. Moreover, 3DVidGen does not present or evaluate 3D geometry quality, making its 3D geometry generation ability unclear, which hinders its applicability in the scenarios requiring good 3D geometry (Yuan et al., 2022). In contrast, our PV3D aims to synthesize realistic videos with high-quality detailed geometry.

# 3 PV3D: 3D PORTRAIT VIDEO GENERATION

## 3.1 OVERVIEW

**Problem formulation.** Given a monocular 2D portrait video collection $\mathfrak{D} = \{\boldsymbol{v}_n\}_{n=1}^N$ consisting of $N$ video sequences, the goal of 3D-aware portrait video generation is to learn a generator $\mathcal{G}$ that synthesizes videos given joint conditions of random noise $\mathbf{z}$, camera viewpoint $c$ and timestep $t$ **without** relying on 3D geometry or multi-view supervision.

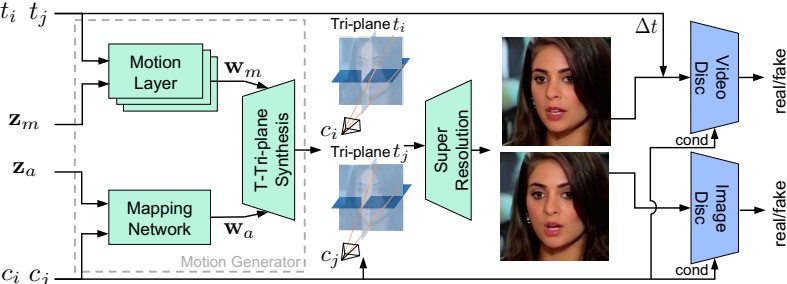

Figure 2: PV3D creates portrait video frames from appearance code $\mathbf{z}_a$, motion code $\mathbf{z}_m$, timesteps $\{t_i, t_j\}$, and camera poses $\{c_i, c_j\}$. Motion layers and mapping network encode inputs into intermediate motion code $\mathbf{w}_m$ and style code $\mathbf{w}_a$, respectively. To incorporate temporal dynamics, the temporal tri-plane synthesis network extends 3D tri-plane (Chan et al., 2022) to spatio-temporal domain. Two camera-conditioned discriminators regularize the image quality and motion plausibility.

**Framework.** The overview of our proposed framework is shown in Figure 2. PV3D formulates 3D-aware portrait video generation task as $\boldsymbol{v} = \mathcal{R}(\mathcal{G}(\mathbf{z}|c, t), c)$ where the generator $\mathcal{G}(\cdot)$ first gen-

erates 3D-aware spatio-temporal representation, followed by volume rendering and super-resolution (denoted as $\mathcal{R}(\cdot)$) to obtain the final video sequence. In this work, our generator $\mathcal{G}$ builds upon tri-plane representation from EG3D (Chan et al., 2022) and extends it to spatio-temporal representation for video synthesis, which we denote as temporal tri-plane. Instead of jointly modeling appearance and motion dynamics within a single latent code $\mathbf{z}$, we factorize the 3D video generation into appearance and motion generation components. Specifically, our PV3D takes two independent latent codes, *i.e.*, appearance code $\mathbf{z}_a \sim \mathcal{N}(\mathbf{0}, \boldsymbol{I})$ and motion code $\mathbf{z}_m \sim \mathcal{N}(\mathbf{0}, \boldsymbol{I})$ as inputs. We condition the generator on $\mathbf{z}_a$ to synthesize varying video appearance, *e.g.*, genders, skin colors, hair styles, glasses, *etc.*, and use $\mathbf{z}_m$ to model motion dynamics, *e.g.*, a person opening his/her mouth.

During training, we randomly sample two timesteps $\{t_i, t_j\}$ and their corresponding camera poses $\{c_i, c_j\}$ for one video. Following EG3D, we project the appearance code $\mathbf{z}_a$ and camera pose $c$ into intermediate appearance code $\mathbf{w}_a$ for content synthesis. As for the motion component, we develop motion layer to encode motion code $\mathbf{z}_m$ and timesteps $\{t_i, t_j\}$ into intermediate motion code $\mathbf{w}_m$. Our temporal tri-plane synthesis network generates tri-plane features based on $\mathbf{w}_a$ and $\mathbf{w}_m$. With the generated tri-plane at $\{t_i, t_j\}$, volume rendering (Max, 1995) is applied to synthesize frames with camera pose $c_i$ and $c_j$, respectively. The rendered frames are then upsampled and refined by a super-resolution module. To ensure the fidelity and plausibility of the generated frame content and motion, we develop two discriminators $\mathcal{D}_{\text{img}}$ and $\mathcal{D}_{\text{vid}}$ to supervise the training of $\mathcal{G}$. Both $\mathcal{D}_{\text{img}}$ and $\mathcal{D}_{\text{vid}}$ are camera-conditioned, which can leverage 3D priors.

### 3.2 3D-AWARE VIDEO GENERATOR

**Challenges.** Existing 3D-aware image GANs can only model static scenes and it remains unclear how to model motion dynamics, such as topological changes of a scene (Park et al., 2022). A straightforward approach is to directly combine the motion condition with latent code, camera pose, and timestep, and feed them to the encoder to generate 3D representations for video rendering. However, such a naive design cannot generate temporally consistent and motion-plausible portrait videos because the generation of video content and motion is highly entangled. Besides, learning 3D-aware portrait video appearance and geometry from monocular 2D videos only is highly under-constrained, making the model training difficult and generation quality poor. Another challenge is that motions caused by camera pose changes (*e.g.*, camera movements) and head pose changes (*e.g.*, look up, turn left) are highly entangled. This introduces ambiguities to the training process, largely increasing model's learning difficulties.

Prior works typically incorporate motion features by either manipulating the latent code for pre-trained image generator (Tian et al., 2021) or simply conditioning 3D representation on latent code and timestep (Bahmani et al., 2022). Nevertheless, such designs cannot guarantee temporal consistency and motion diversity. To address these challenges, we decouple the latent code into appearance and motion components and propose a motion generator to model temporal dynamics. Such design not only preserves the high-fidelity and multi-view consistency of each frame but also enables the synthesis of videos with temporal coherence and motion diversity. Secondly, we propose a camera conditioning strategy to alleviate the motion ambiguity issue and thus facilitates convergence.

**Synthesis of motion dynamics.** To generate motion at each timestep in a video, we condition the generator on latent codes $\mathbf{z}_a$ and $\mathbf{z}_m$, timesteps $\{t_i, t_j\}$ and camera poses $\{c_i, c_j\}$. Without loss of generality, we only describe the generation process for timestep $t_i$ and camera pose $c_i$. As shown in Figure 3, we introduce $K$ motion layers into the synthesis layers of motion generator. Each motion layer encodes motion code $\mathbf{z}_m$ and timestep $t_i$ into intermediate motion code $\mathbf{w}_m^{i,k}$. In particular, the motion code is first multiplied with timestep to encode temporal information, followed by a lightweight mapping head $\mathcal{H}_m$ with leaky ReLU activation. A multi-layer perceptron (MLP) then encodes them into $\mathbf{w}_m^{i,k}$. In other words, the $k$-th motion layer computes

$$\mathbf{w}_m^{i,k} = \text{MLP}_k(\mathcal{H}_m^k(\mathbf{z}_m * t_i)), \tag{1}$$

where $k \in \{0, 1, ..., K\}$. $i \in \{0, 1, ..., N\}$ denotes the frame index while $N$ represents the total number of frames in one video. These intermediate motion codes $\mathbf{w}_m^{i,k}$ are then passed to the temporal tri-plane synthesis network to modulate the static appearance features via adaptive instance normalization (AdaIN) (Karras et al., 2019) to incorporate temporal dynamics.

In practice, we employ an equivalent operator, *i.e.*, modulated convolution (Karras et al., 2020), to compute motion features. We then fuse it with the appearance features controlled by $\mathbf{w}_a^i$. The

fused features are passed to the next synthesis layer iteratively to generate tri-planes. This process is formulated as

$$f_k = \mathcal{S}_k^1(f^* + \text{ModConv}(f^*, \mathbf{w}_m^{i,k}), \mathbf{w}_a^i), \text{ where } f^* = S_k^0(f_{k-1}, \mathbf{w}_a^i). \tag{2}$$

Here, $\mathcal{S}_k^0$ and $\mathcal{S}_k^1$ denote the first and second synthesis block ($\text{ModConv}$) in $k$-th synthesis layer, while $f_k$ denotes the feature map synthesized by the $k$-th layer.

**Motion diversity and temporal consistency.** Recent works (Tov et al., 2021; Shen et al., 2020; Richardson et al., 2021) investigate the semantic meaning of intermediate latent code space ($\mathcal{W}^+$) in pre-trained StyleGAN model, and discover that one can perform diverse manipulation of image content by using different style codes at different style-modulation layers. We made similar observation for PV3D (Appendix A.2). Specifically, mixing latent codes in shallow layers only brings coarse level changes on facial attributes, such as expressions, hair, *etc*, which reduces the diversity. On the other hand, mixing style code in deeper layers leads to drastic changes in image content. As a result, identities will no longer be preserved, causing severe temporal inconsistency.

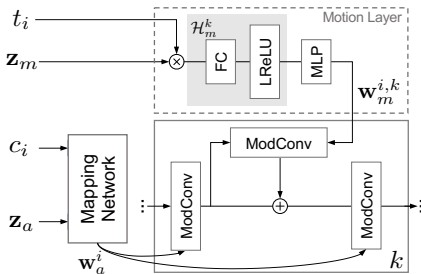

To preserve the identity in generated videos, we carefully select $K$ synthesis layers, such that $k \leq K$, for incorporating motion features. Choosing a suitable $K$ increases temporal consistency and improves our motion generator's capacity for modeling diverse motions. Besides, this synthesis layer selection also alleviates the overfitting of RGB video frames, which improves the quality of 3D geometry.

Figure 3: Architecture of the $k$-th synthesis layer in motion generator. Motion layer encodes $\mathbf{z}_m$ and $t_i$ into intermediate motion code $\mathbf{w}_m^{i,k}$. Motion features are computed by modulating appearance features which are conditioned on $\mathbf{w}_a^i$.

**Alleviating motion ambiguities by camera sequence condition.** Learning 3D-aware portrait video generation from 2D videos faces another challenge, *i.e.*, the entangled face motions and camera movements bring motion ambiguities. The concurrent work 3DVidGen (Bahmani et al., 2022) simply shares one camera pose for the entire video. However, this approach ignores motion ambiguity that harms video fidelity and geometry quality. Differently, we condition the generator on pre-estimated camera pose sequences (Appendix A.3). It has two advantages. 1) Suppose that we are only interested in the faces, a camera rotating around a static face is equivalent to rotating the face in front of a static camera. Generating head rotation directly is challenging due to the large topological changes of the scene. By conditioning our generator on $c_i$ at each time instant $t_i$, we can model the head rotation by rotating camera for observation instead of deforming the 3D scene, making our model easy to optimize. 2) Given the camera pose sequence, our generator can encode view-dependent features for each frame to leverage 3D priors. Such a simple design effectively improves the multi-view consistency and facilitates the learning of dynamics in 3D geometry as verified in our experiments.

## 3.3 CONDITIONAL DISCRIMINATORS

Due to the lack of regularization in video generation procedure, the generated videos may present implausible contents or unreasonable motions. To ensure the plausibility of portrait video generation, we introduce a discriminator module to guide the generation process. In particular, the module consists of an image discriminator $\mathcal{D}_{\text{img}}$ for evaluating video appearance quality and a video discriminator $\mathcal{D}_{\text{vid}}$ for ensuring video motion plausibility.

Our image discriminator $\mathcal{D}_{\text{img}}$ follows EG3D's discriminator architecture and uses camera poses as conditions to guide the generator to learn correct 3D priors and thus it can produce multi-view consistent portraits. We apply $\mathcal{D}_{\text{img}}$ on each generated frame $I_i$ (at timestep $t_i$) independently, which can be formulated as $p_{\text{img}} = \mathcal{D}_{\text{img}}(I_i, c_i)$, where $p_{\text{img}}$ denotes the real/fake probability.

Our PV3D generates two images $\{I_i, I_j\}$ jointly at two random timesteps $\{t_i, t_j\}$ for each video during training, we thus design a camera-conditioned dual-frame video discriminator $\mathcal{D}_{\text{vid}}$ to facilitate motion-plausible portrait video generation. Specifically, we first concatenate the generated two frames $\{I_i, I_j\}$ channel-wisely to obtain an image pair. To help encode temporal information, we

further concatenate the timestep difference $\Delta t = t_j - t_i$ with the image pair. Our video discriminator learns to differentiate the real and generated image pairs based on motion features extracted from this hybrid input. To alleviate motion ambiguity and model view-dependent effects, we further condition $\mathcal{D}_{\text{vid}}$ on the corresponding camera poses $\{c_i, c_j\}$. Our video discriminator is formulated as $p_{\text{vid}} = \mathcal{D}_{\text{vid}}([I_i, I_j, \Delta t], [c_i, c_j])$, where $p_{\text{vid}}$ indicates the probability of each image pair being sampled from real data distribution. Although $\mathcal{D}_{\text{vid}}$ only takes two frames as inputs, it can effectively learn ordinal information (Appendix A.5) and help produce motion-plausible results as verified in our experiments (Section 4.3). Moreover, such a simple design largely improves training efficiency and stability compared with previous methods that take long sequences as conditions (Tulyakov et al., 2018; Tian et al., 2021).

## 3.4 TRAINING AND INFERENCE

**Training.** To synthesize an image $I \in \mathbb{R}^{H \times W \times 3}$ based on tri-plane $\mathcal{T}$, we shoot rays $r(s) = o + sd$ out from the camera origin $o$ along direction $d$ at each pixel (Mildenhall et al., 2020). In practice, we sample query points $x_r$ along each ray and get the features for each point by interpolating them in $\mathcal{T}$. The features are passed to the decoder to predict color $\mathbf{c}$ and density $\sigma$ such that $[\sigma(\mathbf{r}(s)), \mathbf{c}(\mathbf{r}(s))] = \text{Decoder}(\text{Interp}(x_r, \mathcal{T}))$, where Decoder is an MLP with softplus activation, Interp denotes interpolation. The pixel value is calculated by volume rendering as:

$$I(\mathbf{r}) = \int_{s_n}^{s_f} p(s)\sigma(\mathbf{r}(s))\mathbf{c}(\mathbf{r}(s))\mathrm{d}s, \text{ where } p(t) = \exp\left(-\int_{s_n}^{s} \sigma(\mathbf{r}(s))\mathrm{d}s\right). \tag{3}$$

We compute the non-saturating GAN loss (Goodfellow et al., 2020) $\mathcal{L}_{\text{img}}$ and $\mathcal{L}_{\text{vid}}$ and R1 regularization loss (Mescheder et al., 2018) $L_{\text{R1}}$. Following EG3D, we use dual image discriminator which takes both the low-resolution raw image and high-resolution image as inputs. We also compute density regularization $L_\sigma$ on the generated video frames. The overall loss is formulated as:

$$\mathcal{L}_{\text{adv}}^{\mathcal{G}} = \mathcal{L}_{\text{img}}^{\mathcal{G}} + \mathcal{L}_{\text{vid}}^{\mathcal{G}} + \mathcal{L}_\sigma, \mathcal{L}_{\text{adv}}^{\mathcal{D}} = \mathcal{L}_{\text{img}}^{\mathcal{D}} + \mathcal{L}_{\text{vid}}^{\mathcal{D}} + \mathcal{L}_{\text{R1}}. \tag{4}$$

**Inference.** Although trained on sparse frames only, our generator can synthesize contiguous frames during inference. For each video, we generate frames at timestep $t_i$, where $i \in \{0, 1, ..., N\}$, $N$ denotes the maximum number of frames. The mapping network in our generator takes camera pose $c_i$ for each frame to generate intermediate appearance code $\mathbf{w}_a^i$. However, during training, each frame has its camera pose. This brings discrepancy in intermediate appearance codes within the video, which harms the temporal consistency. We propose to share the same $c_i$ for the mapping network in inference, which largely improves temporal consistency as demonstrated in experiments.

## 4 EXPERIMENTS

We study the following questions in our experiments. 1) Can PV3D generate high-quality portrait videos with dynamic 3D geometry and multi-view consistency? 2) How does each component in our PV3D model take effect? 3) What are the performance of PV3D in downstream applications? To answer these, we conduct extensive experiments on several human portrait video datasets.

## 4.1 DATASETS

We experiment on three face video datasets, *i.e.*, **VoxCeleb** (Nagrani et al., 2017; Chung et al., 2018), **CelebV-HQ** (Zhu et al., 2022), and **TalkingHead-1KH** (Wang et al., 2021a). These datasets contain talking face clips of different identities extracted from online videos. We balance the video clips for each identity and preprocess the videos using a standard pipeline (Appendix A.3).

## 4.2 COMPARISONS

**Baselines.** We compare PV3D against four state-of-the-art methods for 3D-aware video generation: (1) the concurrent work **3DVidGen** (Bahmani et al., 2022); (2) **StyleNeRF+MCG-HD**: combining the SOTA 3D image GAN model StyleNeRF (Gu et al., 2022) with the SOTA multi-stage video generation work MoCoGAN-HD (Tian et al., 2021); (3) **EG3D+MCG-HD**: combining EG3D (Chan

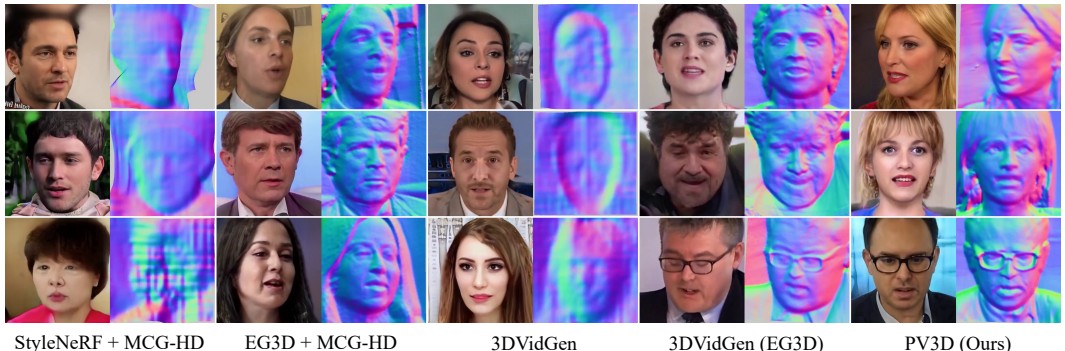

StyleNeRF + MCG-HD     EG3D + MCG-HD     3DVidGen     3DVidGen (EG3D)     PV3D (Ours)

Figure 4: Qualitative comparisons between PV3D and baselines, see our project page for videos.

et al., 2022) with MoCoGAN-HD; (4) **3DVidGen (EG3D)**: replacing the 3D image GAN backbone in 3DVidGen with EG3D.

**Evaluation metrics.** We evaluate PV3D and baseline models by Frechet Video Distance (FVD) (Unterthiner et al., 2018), Multi-view Identity Consistency (ID) (Shi et al., 2021), Chamfer Distance (CD), and Multi-view Image Warping Errors (WE) (Zhang et al., 2022a;b). The evaluation metrics for multi-view consistency are originally proposed for 3D image generation. We extend these metrics to multiple frames which are suitable for 3D video generation tasks. Please refer to Appendix A.4 for more details on our evaluation metrics.

**Quantitative evaluations.** Table 1 summarizes the quantitative comparisons between PV3D and baseline models. First, we observe that PV3D outperforms all of the baseline models on all datasets *w.r.t.* FVD, ID, and WE, which shows our PV3D can generate videos with diverse content and plausible motions while maintaining high multi-view consistency within videos. It is worth noting that our PV3D has a higher CD than StyleNeRF+MCG-HD. This is because CD computes the distance between two point clouds rendered from frontal and side views. As illustrated in Figure 4, StyleNeRF+MCG-HD fails to synthesize detailed 3D geometry, thus leading to lower CD than PV3D. A similar observation is made in TalkingHead-1KH where MCG-HD+EG3D baseline fails to generate videos with diverse motion, which also leads to smaller CD. On the other hand, PV3D is able to generate high-quality *dynamic* 3D geometry with better multi-view consistency.

**Qualitative evaluations.** Figure 4 shows the qualitative comparisons. We first observe that both StyleNeRF+MCG-HD and EG3D+MCG-HD produce poor results because their multi-stage framework design is not end-to-end trainable, resulting in implausible motions. Compared with them, 3DVidGen and its EG3D counterpart achieve relatively better video fidelity. However, the geometry quality and motion diversity of 3DVidGen are still not comparable with PV3D due to their straightforward motion condition design. Differently, our PV3D produces temporally consistent and motion plausible videos with high-quality geometry.

Table 1: Quantitative comparisons, with best results bold and second best underlined.

| | VoxCeleb | | | | CelebV-HQ | | | | TalkingHead-1KH | | | |
|---|---|---|---|---|---|---|---|---|---|---|---|---|
| | FVD↓ | ID↑ | CD↓ | WE↓ | FVD↓ | ID↑ | CD↓ | WE↓ | FVD↓ | ID↑ | CD↓ | WE↓ |
| StyleNeRF+MCG-HD | 348.7 | 0.70 | **1.08** | 36.06 | 134.4 | 0.80 | **1.13** | 38.73 | 292.7 | 0.75 | 5.34 | 49.29 |
| EG3D+MCG-HD | 222.1 | 0.80 | 1.57 | 10.57 | 298.4 | 0.77 | 3.34 | 10.74 | 262.4 | 0.78 | **1.39** | 11.54 |
| 3DVidGen | 65.5 | 0.75 | 3.40 | 44.55 | 63.6 | 0.77 | 3.80 | 37.30 | 83.0 | 0.76 | 4.35 | 46.47 |
| 3DVidGen (EG3D) | 56.3 | 0.71 | 3.65 | 24.55 | 66.2 | 0.70 | 3.83 | 26.34 | 89.8 | 0.65 | 4.56 | 35.48 |
| PV3D (Ours) | **29.1** | **0.81** | 1.34 | **9.76** | **39.3** | **0.81** | 1.21 | **8.18** | **66.6** | **0.80** | 2.33 | **10.73** |

### 4.3 ABLATION STUDIES

In this section, we conduct ablation studies of PV3D on VoxCeleb dataset as it contains more diverse motions and appearances as well as more balanced identity distributions.

**Motion layer position.** PV3D computes and fuses motion features in selected synthesis layers, *i.e.*, the first $K$ layers in our motion generator with a default setting of $K = 4$ (denoted as middle). To study the effect of motion layer position, we insert motion layers until early ($K = 2$) and late ($K$

Table 2: Ablations of PV3D on VoxCeleb. We vary the motion layers, motion generator architecture, camera conditions, and discriminator architectures to study their effects.

| Pos. | FVD↓ | CD↓ | WE↓ | Mot. | FVD↓ | CD↓ | WE↓ | Cam. | FVD↓ | CD↓ | WE↓ | Vid. Dis. | FVD↓ | CD↓ | WE↓ |
|---|---|---|---|---|---|---|---|---|---|---|---|---|---|---|---|
| early | 40.2 | 2.32 | 10.60 | MLP | 41.1 | 1.33 | 9.86 | All | 32.1 | 2.49 | 11.32 | w/o Cam | 34.9 | 4.38 | 10.84 |
| middle | 29.1 | 1.34 | 9.76 | Naive | 36.9 | 1.05 | 10.21 | Non | 55.3 | 3.28 | 23.71 | w/o $\Delta t$ | 38.7 | 1.54 | 10.04 |
| late | 33.3 | 6.24 | 11.56 | $\mathbf{z}_m \to \mathbf{z}_a$ | 37.7 | 1.07 | 9.86 | Map | 38.0 | 1.47 | 10.19 | w/ both | 29.1 | 1.34 | 9.76 |
| learned | 30.5 | 1.81 | 10.31 | Ours | 29.1 | 1.34 | 9.76 | MapT | 29.1 | 1.34 | 9.76 | | | | |

(a) The effect of motion layer positions.

(b) The effect of motion generator architectures.

(c) The effect of camera condition in generator.

(d) The effect of different conditions in video discriminator.

= 7) stages. Moreover, we design a simple learnable parameter for each motion layer and optimize this weight for motion feature fusion (denoted as learned). The results are summarized in Table 2a.

We observe that when the motion layers are too shallow, the model's capacity for modeling dynamics is limited and both video quality and multi-view consistency degrade. However, if we insert motion features into all the synthesis layers (late), FVD is still higher and 3D geometry is largely affected because (1) appearance feature manipulation space is large but hard to optimize, which harms motion plausibility; (2) our training process is highly under-constrained. In this case, the model can easily overfit the RGB frames, which hinders the learning of geometry. Similarly, although the video quality improves when using learnable fusion weights, the multi-view consistency is not as good as reflected by CD and WE metrics because the model still tends to produce better RGB content at the cost of worse geometry due to the lack of 3D supervision.

**Motion generator.** Table 2b summarizes the ablations on our motion generator architectures. Our video generator takes two random codes, *i.e.*, $\mathbf{z}_a$ and $\mathbf{z}_m$. We first study the effect of introducing an independent motion code by replacing motion code $\mathbf{z}_m$ with $\mathbf{z}_a$. It can be observed that although CD reduces by 0.3, all other metrics deteriorate especially FVD, suggesting that motion plausibility is affected when motion code is removed. A similar observation is made in naive implementation. Replacing modulated convolution with simple MLP significantly increases FVD, implying MLP is less effective than modulated convolution in manipulating appearance features.

**Camera conditions.** We investigate how different camera conditioning strategies could affect generation performance. In each training iteration, we sample two video frames along with their camera poses. Thus, we have three options for camera conditioning (sharing camera pose or not) in our generator: (1) **All**: condition the whole generator (both mapping network and rendering) on the shared camera pose; (2) **Non**: condition the whole generator on the camera pose of each frame; (3) **Map**: only share camera pose for mapping network, and use different camera pose to render frames.

As shown in Table 2c, sharing one camera has poor multi-view consistency, with 2.49 CD and 11.32 WE. On the other hand, using non-shared cameras leads to even worse performance. This is because our mapping network takes a camera pose when computing appearance code. Therefore, changing the camera across video frames would bring rapid changes to the appearance code, leading to temporal inconsistency. An alternative is to only share camera poses in the mapping network. However, this strategy works poorly because the camera pose discrepancy in generation process hinders the convergence. On the contrary, sharing the same camera pose in the mapping network only during inference stage (**MapT**) facilitates the training and preserves temporal consistency. Hence, we use **MapT** as our default setting.

**Discriminator conditions.** Table 2d shows the effect of camera pose and time difference conditions in the video discriminator. We can observe that camera conditions in video discriminator can largely improve 3D geometry, while time difference helps improve temporal coherence by encoding auxiliary temporal information.

## 4.4 APPLICATIONS

**Static portrait animation.** Our generator can independently generate video frame at a certain timestep instead of generating from the first frame auto-regressively. This flexible architecture enables static portrait animation. Given the input image and the estimated camera pose, we fix our generator and optimize latent code at timestep $t = 0$. The inversion is performed in $\mathcal{W}^+$ space (Richardson et al., 2021; Abdal et al., 2019). As shown in Figure 5, the GAN inversion based on our generator also produces high-quality 3D shape for the input frame. We then keep the latent code fixed and ran-

domly sample a motion code to drive the portrait with natural motion. With the 3D priors learned by our PV3D, the synthesized videos can also be rendered under arbitrary viewpoints.

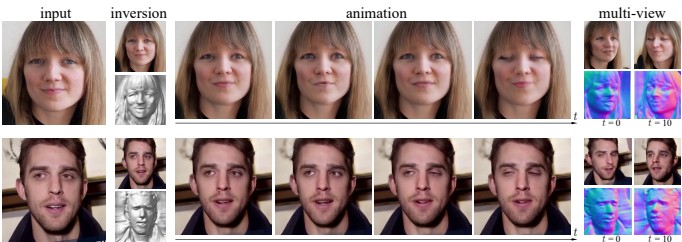

Figure 5: Given static portraits, we fix the video generator and optimize the intermediate appearance code in $\mathcal{W}^+$ space to perform inversion. By sampling a random motion code, we can animate the static portraits with natural motion and synthesize portrait videos with multi-view consistency.

**Monocular video reconstruction and motion editing.** Given a video and its pre-estimated camera pose sequence, we can directly reconstruct the video based on our pretrained generator. For the video content, we also optimize the intermediate appearance code in $\mathcal{W}^+$ space. As for the motion component, we experimentally find that $\mathbf{z}_m^+$ space is more effective. Specifically, we inverse $\mathbf{z}_m$ for each video frame individually. Figure 6 illustrates the results for reconstruction, our PV3D provides a simple solution for 3D reconstruction on monocular videos. Thanks to the disentangled design of motion and appearance components in PV3D, we can fix appearance codes and sample new motion codes to manipulate the motion of input videos in the 3D domain.

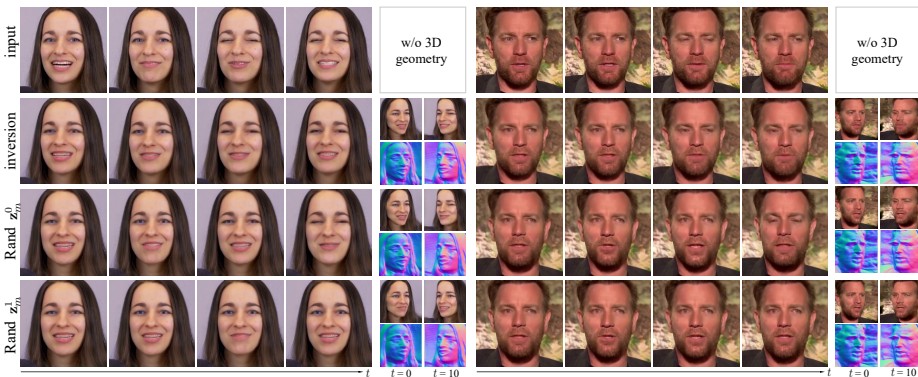

Figure 6: PV3D provides a quick solution for 3D reconstruction on monocular videos. Thanks to the disentanglement of appearance and motion in PV3D, the motion of input videos can be manipulated by changing motion codes. The results can still maintain multi-view consistency.

## 5 CONCLUSION

This work introduces the first 3D-aware generative model, PV3D, for synthesizing multi-view consistent portrait videos with high-quality 3D geometry. By employing independent latent codes for appearance and motion, PV3D can leverage temporal tri-plane synthesis to address the challenges in 3D-aware portrait video generation. Moreover, we condition PV3D on camera pose sequence to alleviate the challenging motion ambiguities. We demonstrate that PV3D can generate both temporal and multi-view consistent portrait videos with diverse motions and dynamic 3D geometry. Besides, PV3D supports downstream applications such as static portrait animation, 3D video reconstruction, and multi-view consistent motion editing. We believe our method will facilitate the desired practical applications in VR/AR and visual effects.

## 6 ACKNOWLEDGEMENTS

This project is supported by the National Research Foundation, Singapore under its NRFF Award NRF-NRFF13-2021-0008 and the Ministry of Education, Singapore, under the Academic Research Fund Tier 1 (FY2022).

## 7 ETHICS STATEMENT

Static portrait animation, monocular video reconstruction, and editing could be misused for generating fake videos or manipulating authentic videos for improper or illegal purposes. These potentially harmful applications may pose a societal threat. We strictly forbid these kinds of abuses. In addition, our video generator may contain bias for face results due to the unbalanced video distribution in training datasets.

## 8 REPRODUCIBILITY STATEMENT

To ensure reproducibility, we describe the implementation details in Appendix. A.1. The dataset preprocessing steps are introduced thoroughly in Appendix. A.3, including video sources, training data balance, face video alignment pipeline, and camera pose estimation approach. Our code and models are publicly available at https://showlab.github.io/pv3d.

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

# A APPENDIX

## A.1 IMPLEMENTATION DETAILS

For each video, we sample two frames within a 16-frame span. Following DiGAN (Yu et al., 2022), we sample timesteps $\{t_i, t_j\}$ from beta distributions. The resolution of the generated video is 512×512. We use a resolution of 64 and a sampling step of 48 for neural rendering during training. In inference stage, we use a rendering resolution of 128 for geometry visualization only. Each camera pose $c$ has 25 dimensions, with 16 for extrinsics and 9 for intrinsics. Our model is implemented using PyTorch. We balance the loss terms by weighting factors: 1) $\lambda_{reg}$=0.6, $\lambda_{vid}$=0.65, $\lambda_{img}$=1.0, $\lambda_{R_1}$=2.0 for VoxCeleb; 2) $\lambda_{reg}$=0.05, $\lambda_{vid}$=0.65, $\lambda_{img}$=1.0, $\lambda_{R_1}$=4.0 for CelebV-HQ; 3) $\lambda_{reg}$=0.5, $\lambda_{vid}$=0.65, $\lambda_{img}$=1.0, $\lambda_{R_1}$=2.0 for TalkingHead-1KH. Our model is trained for 300k iterations with a batch size of 16, which takes 58 hours on 8 Nvidia A100 GPUs.

## A.2 ANALYSIS OF LATENT CODE SPACE

The architecture of synthesis layer in our PV3D largely follows StyleGAN and its following works (Karras et al., 2019; 2020; 2021). Based on the pre-trained StyleGAN models, prior works (Shen et al., 2020) also investigate the property of the latent code space. These works show that the intermediate latent code space has extensive manipulation ability for image synthesis. Although the original design of StyleGAN is sharing one intermediate latent code across all synthesis layers, follow-up works (Abdal et al., 2019; Richardson et al., 2021; Tov et al., 2021) relax this constraint and achieve better reconstruction results for image inversion because using different latent codes for different synthesis layers can further expand the space for image generation.

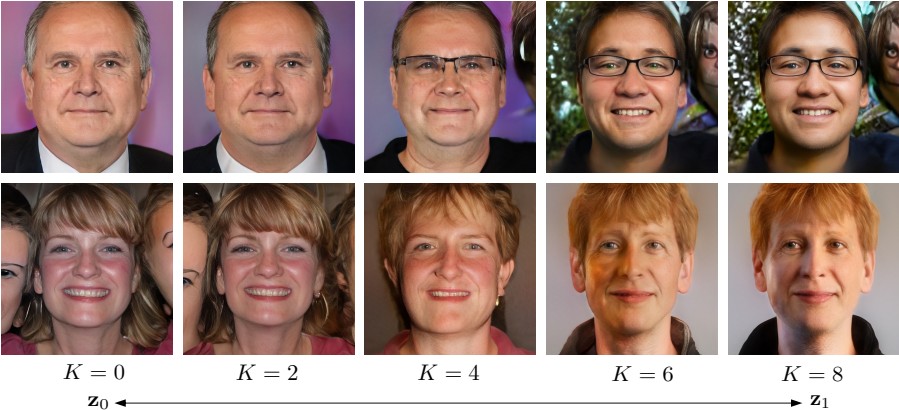

$K = 0$ $\qquad$ $K = 2$ $\qquad$ $K = 4$ $\qquad$ $K = 6$ $\qquad$ $K = 8$

$\mathbf{z}_0 \longleftarrow \qquad\qquad\qquad\qquad \longrightarrow \mathbf{z}_1$

Figure 7: Results of style mixing in first $K$ synthesis layers in pre-trained EG3D. For each example, we first sample one latent code and mix it with another latent code in the first $K$ synthesis layers.

Our PV3D can generate temporal tri-plane features by modulating appearance features based on motion code and timestep. We encode motion code and timestep into intermediate motion codes and then compute motion features in temporal tri-plane synthesis network. Because our synthesis network is built on top of EG3D, we also analyze the latent space to find out how the manipulation of appearance features could affect the synthesis results. As shown in Figure 7, we perform style mixing in the first $K$ synthesis layers of a pre-trained EG3D model. When $K$ increases, the image contents gradually change. Specifically, manipulating the appearance code in $K = 2$ layers can largely preserve the contents. However, only modulating features in the first 2 layers would potentially harm the capacity for content diversity of our video generator. When $K \geq 6$, there exists a sharp change in the image content. Because one important property for portrait video is the temporal coherence, *i.e.*, consistent identity, we finally select $K = 4$ in our motion generator to maintain a good temporal consistency as well as motion diversity.

## A.3 DATASET PREPROCESSING

**Video sources.**

*VoxCeleb* (Nagrani et al., 2017; Chung et al., 2018) is an audio-visual speaker verification dataset containing interview videos for more than 7,000 speakers. It provides speaker labels for each video clip. For each speaker, we sample two video clips that have the highest video resolutions.

*CelebV-HQ* (Zhu et al., 2022) is a large-scale face video dataset that provides high-quality video clips involving 15,653 identities. Compared with VoxCeleb, it contains diverse lighting conditions.

*TalkingHead-1KH* (Wang et al., 2021a) consists of talking head videos extracted from 2,900 long video conferences.

For all of the datasets, we directly download videos with the highest possible resolution from YouTube using the provided uid list.

**Training data balance.** CelebV-HQ and TalkingHead-1KH have unbalanced number of video clips for each identity. To balance the video clips, we perform face clustering to predict pseudo-identities and sample video clips for each identity. Specifically, we extract facial features using pre-trained Arc-Face (Deng et al., 2019a) and employ agglomerative hierarchical clustering (Day & Edelsbrunner, 1984) to predict pseudo-identities and sample at most two videos for each identity.

**Alignment.** All datasets are preprocessed using the same alignment pipeline: We first detect landmarks for each frame using an off-the-shelf face alignment package (Bulat & Tzimiropoulos, 2017). Such single frame face alignment technique would introduce temporal inconsistency. Following (Fox et al., 2021), we use a low-pass Gaussian filter to smooth the estimated keypoints before warping the images. Then, we follow the image warping approach of FFHQ (Karras et al., 2019) to align each frame. To facilitate the training of unconditional 3D image GAN, EG3D employs an extra cropping step to process the image. Specifically, they realign the image in depth direction, which forces all of the keypoint of nose to the same point in the world coordinate defined by the parametric face model, *i.e.*, 3DMM (Paysan et al., 2009). We follow this step to process the video clips. Finally, we apply deep face reconstruction (Deng et al., 2019b) to estimate camera pose for each video frame. Again, this process brings temporal inconsistency and we also apply the low-pass Gaussian filter to smooth the results.

**Verification.** Our preprocessing pipeline is automated and purely based on off-the-shelf packages. However, there exist noise and failure cases in each preprocessing step. Therefore, we apply an extra verification step to remove the noisy video clips. In particular, we use ArcFace to extract the facial features again for every 2 frames within a video clip. If the similarity scores between one frame and others are below a threshold $\tau = 0.5$, this frame will be labeled as noisy. We discard video clips that contain more than two noisy frames.

## A.4 EVALUATION METRICS

**Frechet Video Distance (FVD).** We compute the statistics for ground-truth and generated samples using the pre-trained I3D (Carreira & Zisserman, 2017) model (the PyTorch version checkpoint released by VideoGPT[1]). For the training dataset, we randomly sample 5000 videos. Each video generator synthesizes 5000 uncurated videos randomly for test. All of the videos are encoded in H264 format. Compared with previous methods that save videos in PNG format (Skorokhodov et al., 2022), we empirically find that saving videos with H264 and decoding videos into images during testing does not cause any variation and can largely save space.

**Multi-view Identity Consistency (ID).** 3D-aware image GAN works compute ID by rendering both frontal view and side view images to measure the model's multi-view identity consistency. They adopt state-of-the-art face recognition model (Deng et al., 2019a) (denoted as $\mathcal{F}$) to extract features and compute similarity scores between frontal and side view images. We simply extend this process to multiple frames. In our experiments, we generate random videos and render both frontal face and side face images $I_y$, at two randomly sampled timesteps $\{t_0, t_1\}$, where y denotes yaw angles. The ID metric is formulated as:

$$\text{ID}(I_{y_0}^{t_0}, I_{y_1}^{t_1}) = \mathcal{F}(I_{y_0}^{t_0})^T \mathcal{F}(I_{y_1}^{t_1}), \tag{5}$$

---

[1]https://github.com/wilson1yan/VideoGPT

where $t_0 \neq t_1$ or $y_0 \neq y_1$. In this work, we report results with $y \in \{0°, 30°\}$ on 1000 videos. We compute ID for each frame pair within each video. The final ID metric is the average value for all of the image pairs.

**Chamfer Distance (CD).** StyleSDF (Or-El et al., 2022) proposes to use the Chamfer Distance between the frontal and side view point clouds to measure the multi-view consistency of 3D geometry. In this work, we extend this metric to multiple frames by computing CD between depth map pairs in each video. For each video, we sample two timesteps $\{t_0, t_1\}$ and render point cloud at two angles $P_y$, where y denotes yaw angles. The CD is mathematically formulated as:

$$\text{CD}\left(P_{y_0}^{t_0}, P_{y_1}^{t_1}\right) = \underset{x \in P_{y_0}^{t_0}}{\text{med}} \min_{y \in P_{y_1}^{t_1}} \|x - y\|_2^2 + \underset{y \in P_{y_1}^{t_1}}{\text{med}} \min_{x \in P_{y_0}^{t_0}} \|x - y\|_2^2, \tag{6}$$

where med means median, $t_0 \neq t_1$ or $y_0 \neq y_1$. Following StyleSDF we normalize the point clouds based on the volume sampling bin size for each generator before computing CD. We also remove the non-terminating rays whose opacity is below 0.5 for all of the models whose backbone is EG3D. To make fair comparisons, for baselines based on StyleNeRF, we render point clouds in foreground NeRF, *i.e.*, rendering face part only. We interpolate all of the points clouds to 64×64. We also use y $\in \{0°, 30°\}$ and render 1000 videos. We then average CD for all of the point cloud pairs to get the final CD result.

**Multi-view Image Warping Errors (WE).** Inspired by recent 3D image GAN works (Zhang et al., 2022b;a), we can further compute the warping error by reprojecting each pixel from side view to the frontal view based on the images $(I_{y_0}^{t_0}, I_{y_1}^{t_1})$ and depth maps $(D_{y_0}^{t_0}, D_{y_1}^{t_1})$ at yaw angle $\{y_0, y_1\}$ and timesteps $\{t_0, t_1\}$. The camera extrinsics and intrinsics are [R|t] and K, where R is determined by yaw angle y. We warp a pixel located at $(i, j)$ (denoted as $\mathbf{x}$) in side view image to frontal view by re-projection. It is formulated as:

$$\mathbf{x}' = \text{K}[\text{R}'|\text{t}'][\text{R}|\text{t}]^{-1}[\text{K}^{-1}\mathbf{x}, D(\mathbf{x})]^T. \tag{7}$$

Based on the re-projected coordinates, we compute the warping error as:

$$\text{WE}(I_{y_0}^{t_0}, I_{y_1}^{t_1}) = \frac{1}{L} \sum |I_{y_0}^{t_0} - I_{\text{warp}}|, \tag{8}$$

where $I_{\text{warp}}(\mathbf{x}') = I_{y_1}(\mathbf{x})$ and $y_0 \neq y_1$, $L$ is the number of re-projected points that located within the field-of-view in the frontal view image, and we only compute WE based on these visible pixels. Both image and point cloud are resized to 256×256. Finally, we use y $\in \{0°, 30°\}$ and generate 1000 videos. We compute the warping error for each front and side view pair in one video. The WE reported in this work is the average value of all the pairs.

## A.5 ADDITIONAL EXPERIMENT RESULTS

**Ablations.** We perform additional ablation study on PV3D by varying the design of *discriminator components* and *volume rendering resolutions*.

Table 3: Ablations on discriminators, resolution for neural rendering, and generator camera pose conditions.

| Disc. | FVD↓ | CD↓ | WE↓ |
|---|---|---|---|
| w/o Vid | 145.9 | 3.89 | 11.82 |
| w/o Img | 42.2 | 3.44 | 14.40 |
| w/ both disc. | 29.1 | 1.34 | 9.76 |

(a) The effect of discriminator components.

| Rend. Res. | FVD↓ | CD↓ | WE↓ |
|---|---|---|---|
| 32 | 42.4 | 3.25 | 12.98 |
| 64 | 29.1 | 1.34 | 9.76 |
| 128 | 34.1 | 0.92 | 9.62 |

(b) The effect of volume rendering resolution.

| Gen. Cam. | FVD↓ | CD↓ | WE↓ |
|---|---|---|---|
| w/o cam | 44.3 | 2.35 | 13.01 |
| w/ cam | 29.1 | 1.34 | 9.76 |

(c) The effect of camera conditioning in generator.

*Discriminator components.* PV3D employs two independent discriminators to regularize the video content and motions. We ablate the image and video discriminators to study how each one supervises the training of generator. Table 3a illustrates the results of removing each discriminator. Without video discriminator, the video quality largely deteriorates, suggesting that video discriminator is able to guarantee motion plausibility. In addition, removing image discriminator also brings

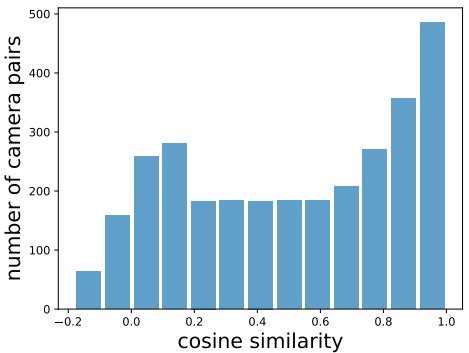

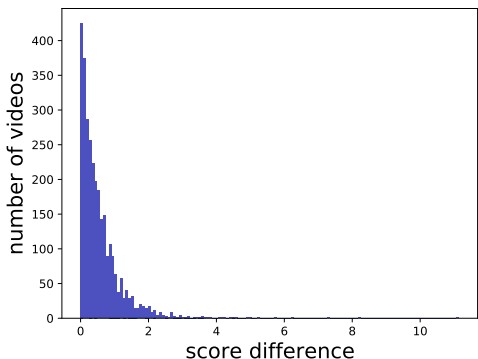

(a) Statistics for cosine similarities between the embeddings of original and flipped camera pose pairs.

(b) Statistics for real/fake score differences between original and flipped camera pose pairs.

Figure 8: The analysis for the alignment ability of our video discriminator. We sample 3000 camera pose sequences from training dataset and input both original camera pose sequences together with the reversed ones into our video discriminator to study its ability for encoding ordinal information.

a significant performance drop in FVD, CD, and WE, which proves that our video discriminator can also supervise the frame content but struggle to guarantee 3D geometry.

*Volume rendering resolution.* Table 3b summarizes the results of using different resolutions for neural volume rendering during training. A small resolution (32) restricts model's capacity, leading to worse performance on all of the metrics. Although a higher resolution (128) outperforms our default setting (64) in terms of multi-view consistency and warping error, its video quality is still worse. The results show that there exists a tradeoff between the video quality and 3D geometry in our generator because it is trained only on 2D videos. Considering the computation resources, our default setting uses a resolution of 64. In this work, we also use 64 for testing and reporting evaluation metrics. Moreover, we only use a resolution of 128 for geometry visualization.

*Generator camera conditioning.* PV3D takes camera poses to encode 3D priors in generator. Table 3c shows that the 3D priors improve performance with a large margin. Without the camera pose condition in generator, a large decrease can be observed in all of the evaluation metrics.

**Video discriminator alignment.** To investigate whether our video discriminator can align the camera poses with the input video frames, we conduct two following experiments.

*Effects of camera pose order on embedding.* We sample 3000 camera pairs from the training dataset and pass them to our video discriminator. We then exchange the order of the camera poses and compute the embeddings again. After that, we compute the cosine similarities between the embeddings of original and the flipped camera pose sequences. The statistics for similarities are shown in Figure 8a. It can be observed that most of the camera pose sequences with reversed orders will be mapped into embeddings that are far apart. Thus, our video discriminator is sensitive to the camera pose orders.

*Effects of camera pose order on discrimination ability.* We further input the videos generated by the sampled camera poses together with both original and flipped camera pose sequences into the video discriminator. We compute the L1 distances between the real/fake scores and visualize statistics in Figure 8b. It illustrates that our video discriminator predicts different real/fake scores for most of the video frame pairs if the camera pose sequences are in reverse order. The results prove that our video discriminator is order-aware and can align the camera pose sequence with the video frame pair.

**Longer-term video synthesis.** PV3D is trained on two video frames per clip, this architecture makes it suitable for long-term video synthesis. To study its ability for long-term video generation, we further train and test PV3D on video clips of 48 frames. The results are shown on our project page. As we can see, training data with longer duration contains more diverse motions and PV3D can learn to generate such motions accordingly.

**Qualitative results.** We demonstrate more results of PV3D in Figure 9.

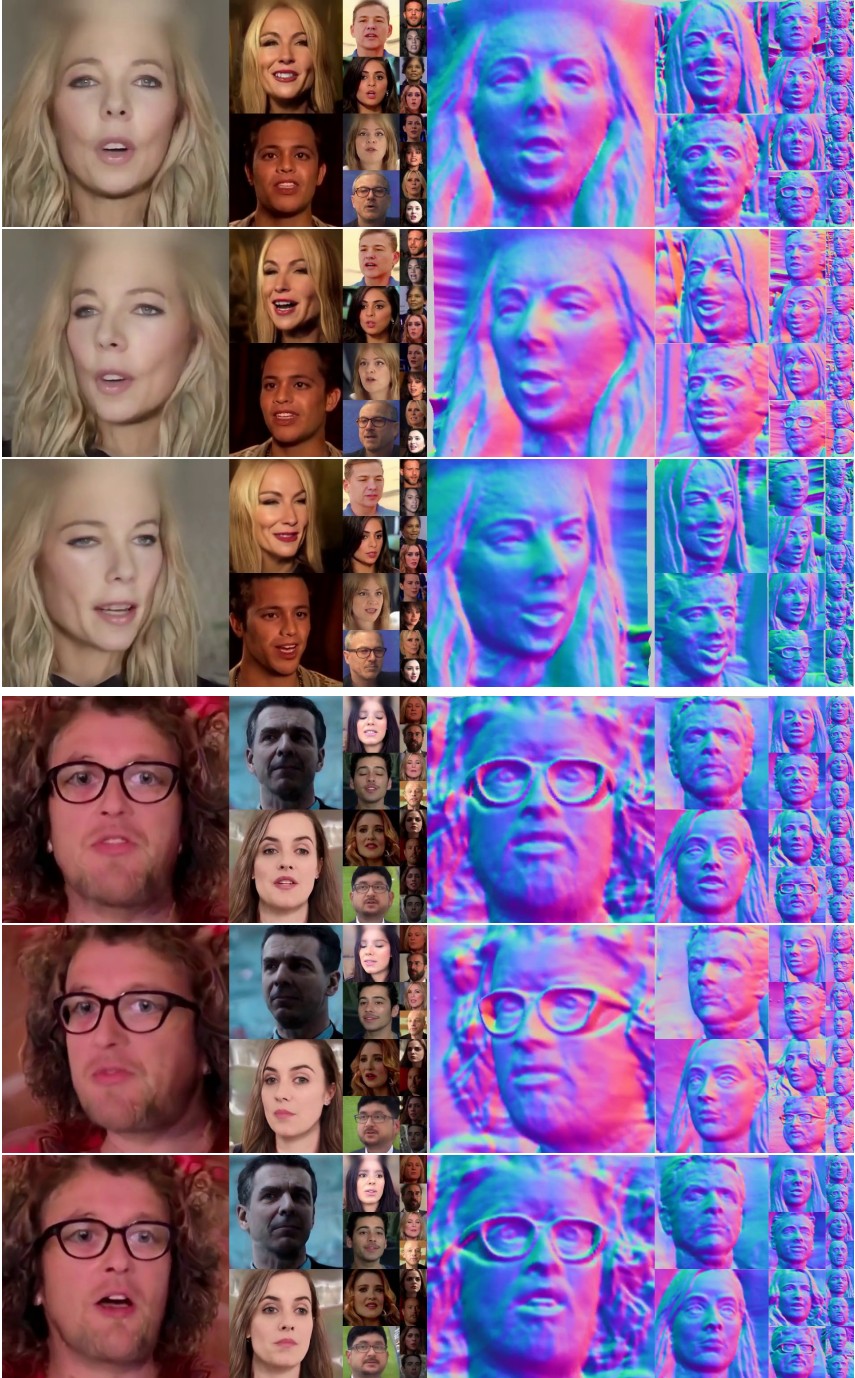

Figure 9: Samples generated by PV3D, see our project page for video results.

## A.6 LIMITATIONS AND FUTURE WORK

Our PV3D has several limitations: 1) PV3D is trained and tested on video clips that contain at most 48 frames. The model's ability for modeling long-term (minutes order) dynamics is unknown. 2) The 2D video dataset quality is not comparable to image datasets such as FFHQ and CelebA. Our model has a flexible architecture that may support pre-training or joint training on image datasets, yet this augmentation strategy has not been explored although it is promising and meaningful. For future work, we will explore modeling long-term dynamics with novel 3D representations that are more suitable for 3D video generation, and leverage high-quality image datasets for data augmentation.

