# OpenReview forum: "PV3D: A 3D Generative Model for Portrait Video Generation"
_ICLR.cc/2023/Conference — ICLR 2023 poster_

### Official Review · Reviewer_YSuv · 2022-10-22

**Confidence:** 4
**Correctness:** 3
**Technical Novelty And Significance:** 3
**Empirical Novelty And Significance:** 3
**Recommendation:** 8

**Clarity, Quality, Novelty And Reproducibility:**

The writing is clear, and the results are of good quality. While the method uses existing EG3D architecture and two discriminators that are commonly used in video generation, I believe the work has its contributions to the community. Given that EG3D code is released, I don’t worry about reproducibility.


**Strength And Weaknesses:**

Strength:
- The motivation of the proposed paper is interesting. The paper is well-written. The video quality looks reasonable.

Weaknesses:
- It’s unclear in the paper how is this work different from [1]. Please clarify.
- I doubt if the video discriminator that only takes paired frames w/ the corresponding time delta is sufficient to maintain spatio-temporal consistency, especially for texture and albedo. One example is the man on the left in the first video. His mustache flickered over time, i.e., it looks faded sometimes. Also, from other generated portrait videos, I can see that the head shape deformation over time as it dynamically became bigger and smaller. Similar inconsistency happens to hair.
- Following the last question w.r.t. spatio-temporal consistency, could the authors generate some videos from multiple views at one time for the same person, e.g., rendering 3 videos with the fixed left, middle, and right views, simultaneously?
- It's unclear how the same motion features work on various face identities. To verify the constant effectiveness of motion conditioning, could the authors show multiple image/video results (i.e., different face identities) with the same motion features?
- Can this work generate face videos with complex expressions, which the earlier work [1] can handle?
- I’m worried about data processing. The authors used alignment which is commonly used in image preprocessing but it might not be proper for videos. Usually, people crop videos with a fixed cropping window. If doing alignment, will the processed videos become jittering? Did the authors observe this issue and fix it? Please explain.


**Summary Of The Paper:**

The paper proposes a generative framework for synthesizing multi-view consistent portrait videos. Towards this, the authors employed EG3D architecture in NeRF and made the network conditioned on motion features. 2 discriminators are used to maintain spatial and temporal consistency. While it’s claimed in the paper that this is the first work for the task, I have noticed the existing method [1] doing the same task and they are able to generate 3D-aware portrait videos with motion controllability.

[1] P. Zhuang et al., Controllable Radiance Fields for Dynamic Face Synthesis, 3DV 2022.


**Summary Of The Review:**

The paper is well-structured and the topic is interesting. The video quality is good. I’ll wait for the authors to reply to the questions.

---

> ### Author Response · Authors · 2022-11-15
> **Response to reviewer YSuv (2/2)**
>
> > **4. It's unclear how the same motion features work on various face identities. To verify the constant effectiveness of motion conditioning, could the authors show multiple image/video results (i.e., different face identities) with the same motion features?**
>
> We have rendered the videos using various appearance codes and the same motion code. The results can be found on our project page (the 5X5 video grid under caption “Motion Retargeting”). As we can see, the results are reasonable although motion is not exactly identical. Since our generator first synthesizes static features based on appearance code, the motion features will thus be slightly affected by the appearance code.
>
> > **5. Can this work generate face videos with complex expressions, which the earlier work [1] can handle?**
>
> Since our work is a pure unconditional GAN model, it cannot provide an explicit control of the expression. Thus, the ability of generating videos with complex expressions depends on whether the training data contains such complex expressions. We visualize some examples on our project page (the video grid with caption “Portrait Videos Samples”). These examples show that our model can still synthesize portrait videos with complex expressions. For example, in video (f), the person starts from an exaggerated expression and the wrinkles in her forehead change later. Video (b) also starts with a large opening mouth. Video (a) and (d) contain photo-realistic gaze changes, while video (c) contains continuous large head pose changes. Compared with our model, [1] can handle complex expressions because it takes pre-estimated motion (expression) features as input and also computes motion reconstruction loss. In our future work, we will explore the conditional PV3D model to enable the explicit control for the 3D portrait video generation.
>
> > **6. I’m worried about data processing. The authors used alignment which is commonly used in image preprocessing but it might not be proper for videos. Usually, people crop videos with a fixed cropping window. If doing alignment, will the processed videos become jittering? Did the authors observe this issue and fix it? Please explain.**
>
> As we mentioned in point 4, we do observe noises, especially temporal inconsistency, in the data processing steps. To alleviate this, we follow prior work [4] to smoothen the estimated sequences by employing low-pass Gaussian filters. We apply the filters in all of our dataset preprocessing steps to suppress noises which may cause jittering. Please refer to our Appendix Sec A.3 for more details. With suitable filter parameters, our processed videos are much smoother and jittering is significantly reduced though it cannot be completely eliminated.
>
> **References**
>
> [1] P. Zhuang et al., Controllable Radiance Fields for Dynamic Face Synthesis, 3DV 2022.
>
> [2] I. Skorokhodov, et. al. Stylegan-v: A continuous video generator with the price, image quality and perks of stylegan2. CVPR 2022.
>
> [3] S. Yu, et al. Generating videos with dynamics-aware implicit generative adversarial networks. ICLR 2022.
>
> [4] F., Gereon, et. al. StyleVideoGAN: A Temporal Generative Model using a Pretrained StyleGAN. BMVC 2021.

---

> ### Author Response · Authors · 2022-11-15
> **Response to reviewer YSuv (1/2)**
>
> We thank the reviewer for the insightful and helpful feedback and agree that 1) our work is interesting; 2) our results are reasonable. We respond to each of your comments one-by-one in what follows. Visualization results can be found on our project page https://iclr2023-pv3d.github.io.
>
> > **1. It’s unclear in the paper how is this work different from [1]. Please clarify.**
>
> The differences between our work and [1] are:
> - The tasks are slightly different.  [1] proposes a conditional GAN model which takes the motion features that are pre-computed offline. Thus, [1] first compare their model with FOMM and other image animation works. While our generator is a pure unconditional GAN model which learns to synthesize portrait videos from random noises. Our model does not condition on pre-estimated motion features or expression features.
>
> - [1] and ours focus on different aspects. From [1]’s experiments, we can see that [1] evaluates the model using FID, FVD, identity preservation, and exp matrices. Thus, [1] mainly focuses on the synthesized image/video RGB content. While our work does not only focus on RGB frames, but also evaluates the multi-view consistency and 3D geometry quality. As we claimed, our work is the first 3D-aware GAN for generating portrait videos with high-quality 3D geometry.
>
> - There are many differences in the pipeline, e.g., [1] adds the style code with the motion code predicted by a mapping network whereas ours contains a motion network, which modulates the static features using motion code directly. Moreover, our framework is only supervised by two discriminators, while [1] also applies spatio-temporal consistency loss, identity consistency loss, and background loss, which is much more complicated.
>
> > **2.1 I doubt if the video discriminator that only takes paired frames w/ the corresponding time delta is sufficient to maintain spatio-temporal consistency, especially for texture and albedo. One example is the man on the left in the first video. His mustache flickered over time, i.e., it looks faded sometimes.**
>
> To measure the spatio-temporal consistency, we compute the MSE error between adjacent frames of the generated videos. We randomly render 5000 videos using each model trained on VoxCeleb dataset and report the average MSE errors. The results are summarized in the below table.
>
> |     | StyleNerf+MCG-HD | EG3D+MCG-HD | 3DVidGen | 3DVidGen(EG3D) | PV3D(Ours) |
> |-----|:------------------:|:-------------:|:----------:|:----------------:|:------------:|
> | MSE&#8595;| 46.32            | 30.16       | 22.49    | 20.97          | 18.79      |
> |||||||
>
> The table shows that although our video discriminator only takes two frames, it outperforms all of the baseline models and can generate videos with the best spatio-temporal consistency. Using only a few frames to compute video adversarial loss has been successfully applied in several recent works [2,3]. It has been proved effective in increasing video plausibility and reducing the computation cost. Thus, we also adopt this design. We believe that using more frames could be helpful. We will explore this in our future work.
>
> The flickering issue is challenging for GAN models because convolution-based GAN models cannot handle high-frequency regions (e.g. moustache) well. Both single image GAN models (StyleGAN2, EG3D, StyleSDF) and our video generator have this issue. Recent work such as StyleGAN3 employs frequency truncation to alleviate the flickering. Hence, one of our future directions is to explore how to alleviate this issue to further improve the consistency.
>
> > **2.2 Also, from other generated portrait videos, I can see that the head shape deformation over time as it dynamically became bigger and smaller. Similar inconsistency happens to hair.**
>
> The main cause of head shape and hair deformation is the noise introduced by the off-the-shelf facial landmarks detection model. There exists failure and noises in the facial landmark detection and thus face alignment brings deformations into the final videos. Although we smoothen the detected results using Gaussian filters, there still inevitably exists some temporal inconsistency. We believe a more accurate and robust landmark detector can alleviate this issue. Also, another promising solution is to employ video-based landmark detection models to improve the consistency in our video preprocessing pipeline.
>
> > **3. Following the last question w.r.t. spatio-temporal consistency, could the authors generate some videos from multiple views at one time for the same person, e.g., rendering 3 videos with the fixed left, middle, and right views, simultaneously?**
>
> We have rendered several examples, please check our project page for the videos (the video grid with caption “Simultaneous Multi-view Rendering”). We render the videos simultaneously at phi = -40, 0, 40 degrees.

---

### Official Review · Reviewer_zYuC · 2022-10-25

**Confidence:** 3
**Correctness:** 3
**Technical Novelty And Significance:** 3
**Empirical Novelty And Significance:** 3
**Recommendation:** 8

**Clarity, Quality, Novelty And Reproducibility:**

- Clarify: the paper is well-written.
- Quality: the results demonstrate the efficacy of the approach.
- Novelty: the idea is fresh and interesting.
- Reproducibility: authors state that the code will be made public.

**Strength And Weaknesses:**

## Strengths

The proposed idea is interesting. The experiments are thorough and solid. The paper is clearly written.

## Weakness

I do not observe major issues. However, there are some question I hope authors can clarify. Please see below.

## Questions

**Network structure**

It is a little bit unclear to me how the exact structure of the "T-Tri-plane Synthesis" module in Fig. 2 looks like. From Appendix A.2, I feel like it is essentially some top-layers of vanilla StyleGAN2's generator since the first K layers of the original StyleGAN2's generator have been replaced by the "motion layer". Can authors clarify?

**Camera pose conditioning**

1. From the quantitative perspective, it is unclear to me whether conditioning the generator on the camera pose in necessary since there is no ablations about removing the generator's camera conditioning.

2. From Fig. 2: it seems like during training, for a specific time $t_i$, the motion generator, the rendering from tri-plane, and the discriminator take the same camera pose $c_i$. Is this correct?

3. For Sec.4.3's "Camera Conditioning":
    1) For Tab. 2(c): it is unclear what the final strategy PV3D chooses is.
    2) For "All" strategy, I think the description of "condition the whole generator on the shared camera pose" is misleading. At least from Fig. 2, "the whole generator" only has "mapping network" that can take camera pose as conditioning. My guess is that authors also refer to the rendering process. However, it is not in the generator.
    3) For "Map" strategy: can authors clarify how to choose the "shared pose" given two poses?

**Typos**
- Eq. (2): there are unmatched parenthesis

**Lacked references**

- Zhang et al., Controllable Radiance Fields for Dynamic Face Synthesis. 3DV 2022.
- Schwarz et al., VoxGRAF: Fast 3D-Aware Image Synthesis with Sparse Voxel Grids. NeurIPS 2022.
- Zhao et al., Generative multiplane images: making a 2D GAN 3D-aware. ECCV 2022.

**Summary Of The Paper:**

This paper tackles the problem of training 3D generative models for portrait video generation with only 2D videos. In order to enforce temporal consistency and diverse motion, PV3D utilizes disentangled appearance and motion latent space. Extensive experiments demonstrate the effectiveness of the proposed approach.

**Summary Of The Review:**

This paper tackles the important problem of 3D portrait video generator. The proposed approach is interesting and has been thoroughly verified. Therefore, I vote for acceptance.

---

> ### Author Response · Authors · 2022-11-15
> **Response to reviewer zYuC**
>
> We thank the reviewer for the positive feedback and agree that our work is effective and our experiments are thorough. We answer all of your questions one-by-one in what follows.
>
> **Network structure**
>
> > **It is a little bit unclear to me how the exact structure of the "T-Tri-plane Synthesis" module in Fig. 2 looks like. From Appendix A.2, I feel like it is essentially some top-layers of vanilla StyleGAN2's generator since the first K layers of the original StyleGAN2's generator have been replaced by the "motion layer". Can authors clarify?**
>
> Yes, it is correct. Our “T-Tri-plane Synthesis” module in Fig.2 is built on top of StyleGAN2 synthesis blocks. The difference is that we use “motion layer” in the first K blocks. Specifically, “motion layer” encodes motion code and timestep into the intermediate motion code to modulate the synthesized static features. This can condition our synthesis module on the timestep. Finally, our generator backbone can synthesize a tri-plane sequence.
>
> **Camera pose conditioning**
>
> > **1. From the quantitative perspective, it is unclear to me whether conditioning the generator on the camera pose is necessary since there is no ablations about removing the generator's camera conditioning.**
>
> Per request, we conduct additional ablation studies on the camera pose conditioning in the generator. The results are summarized in the below table.
>
> | *Gen. Cam.* | FVD&#8595;| CD&#8595;  | WE&#8595;   |
> |-----------|------|------|-------|
> | w/o cam   | 44.3 | 2.35 | 13.01 |
> | w/cam     | 29.1 | 1.34 | 9.76  |
> ||||
>
> It can be observed that there exists a large performance drop in all the evaluation metrics if the generator does not condition on camera poses. We have added this table into the revised version. Please refer to the highlighted part in Appendix A.5 for more details.
>
> > **2. From Fig. 2: it seems like during training, for a specific time $t_i$, the motion generator, the rendering from tri-plane, and the discriminator take the same camera pose $c_i$. Is this correct?**
>
> Yes, it is true. During training, we found that using the camera pose $c_i$ of each frame at $t_i$ achieves the best results. This design is studied in our ablation study (Sec. 4,3 Table 2(c)).
>
> **3. For Sec.4.3's "Camera Conditioning":**
>
> > **1. For Tab. 2(c): it is unclear what the final strategy PV3D chooses is.**
>
> The defaults setting is **MapT**. We have updated the main text (Sec. 4.3) to explain our default setting (highlighted in blue).
>
> > **2. For "All" strategy, I think the description of "condition the whole generator on the shared camera pose" is misleading. At least from Fig. 2, "the whole generator" only has "mapping network" that can take camera pose as conditioning. My guess is that authors also refer to the rendering process. However, it is not in the generator.**
>
> That is correct, “All” refers to the whole rendering process. We have updated the main text (Sec. 4.3, highlighted in blue).
>
> > **3. For "Map" strategy: can authors clarify how to choose the "shared pose" given two poses?**
>
> We repeat the camera pose of the first frame and then pass them to the mapping network.
>
> **Typos**
>
> > **Eq. (2): there are unmatched parenthesis**
>
> Thanks for pointing this out, we have updated this equation in the revised version (highlighted in blue).
>
> **Lacked references**
>
> - **Zhang et al., Controllable Radiance Fields for Dynamic Face Synthesis. 3DV 2022.**
> - **Schwarz et al., VoxGRAF: Fast 3D-Aware Image Synthesis with Sparse Voxel Grids. NeurIPS 2022.**
> - **Zhao et al., Generative multiplane images: making a 2D GAN 3D-aware. ECCV 2022.**
>
> Thanks for the suggestions, we have included these references in the related work (highlighted in blue).

---

### Official Review · Reviewer_7Zbn · 2022-10-26

**Confidence:** 4
**Correctness:** 4
**Technical Novelty And Significance:** 4
**Empirical Novelty And Significance:** 4
**Recommendation:** 10

**Clarity, Quality, Novelty And Reproducibility:**

The paper is well-written and clear. The technical ideas presented are novel and original. The method is simple, demonstrating that a recurrent network is not necessary for the task.

**Strength And Weaknesses:**

Strengths: These are the first 3D portrait video results. The quality of results are impressive, in terms of geometry, appearance and motion. The paper is well-written, and the evaluations are thorough.

Weaknesses:
- Since the motion component only controls the first few generator layers, fine-grained motion would be difficult to model. In addition, any lighting effects would not be modeled with the first four latents. I suspect the method does not generate any result where the lighting on the face changes between different timesteps.
- As mentioned in the limitations, the method only deals with very small motion (upto 16 frames). It would be good to show some results with large motion, just to understand how it would fail. I suspect that the design choice of using a single motion latent vector for all timesteps is the limiting factor here.
- For inversion, in Fig. 6, a different motion code is optimized for each frame. How different are the optimized codes for different frames? Would it be possible to use these optimized motion codes to retarget a different appearance code, or are they out-of-distribution?

**Summary Of The Paper:**

The paper proposes a 3D portrait video generative model learned from a dataset of monocular videos. A motion latent code is added to a triplane-based 3D GAN, in addition to an appearance code. A single motion latent controls the motion for the entire temporal sequence. The network also receives the timestep and camera poses as conditioning inputs. Temporal and Static discriminators are used for training.

**Summary Of The Review:**

The paper presents an interesting solution to a novel problem with high-quality results. It would be good to better demonstrate the limits of the method in terms of duration of synthesized videos.

---

> ### Author Response · Authors · 2022-11-16
> **Response to reviewer 7Zbn (2/2)**
>
> > **2. As mentioned in the limitations, the method only deals with very small motion (up to 16 frames). It would be good to show some results with large motion, just to understand how it would fail. I suspect that the design choice of using a single motion latent vector for all timesteps is the limiting factor here.**
>
> We experiment on 48-frame video clips (x3 times longer) of VoxCeleb dataset. The quantitative results are shown in the table below.  To study the performance, we also train a baseline model 3DVidGen on 48 frames and make comparisons between ours and 3DVidGen.
> |*Frame 48*| FVD&#8595;  | ID&#8593;  | CD&#8595;   | WE&#8595;   |
> |------------|-------|------|------|-------|
> | 3DVidGen   | 247.5 | 0.77 | 3.06 | 40.97 |
> | PV3D(Ours) | 152.8 | 0.80 | 1.77 | 11.44 |
> ||||||
>
> The qualitative results can be found on our [project page](https://iclr2023-pv3d.github.io/) (video grid with caption "Long video synthesis"). It shows that our framework can synthesize reasonable long portrait videos.
>
> To investigate whether using a single motion vector is the limiting factor or not. We replace the single motion vector with multiple vectors. The comparisons are shown in the below table. As we can see, using multiple frames causes significant performance drop in FVD and CD. The generated videos suffer from serious temporal inconsistency, i.e., jittering.
>
> | $z_m$        | FVD&#8595;   | ID&#8593;   | CD&#8595;    | WE&#8595;    |
> |--------------|-------|------|-------|-------|
> | Multi        | 182.9 | 0.77 | 11.56 | 13.91 |
> | Single(Ours) | 152.8 | 0.80 | 1.77  | 11.44 |
> ||||||
>
> > **3. For inversion, in Fig. 6, a different motion code is optimized for each frame. How different are the optimized codes for different frames? Would it be possible to use these optimized motion codes to retarget a different appearance code, or are they out-of-distribution**
>
> We tried different designs for video inversion based on our model. We found that optimizing only one motion vector for the entire video cannot inverse the motion component well. We think this is due to the scale of training datasets. The VoxCeleb dataset only contains 14k videos of 7k identities which is not comparable with image datasets such as FFHQ that has 70k images of different people. Thus, we borrow the idea from single image inversion and use the expanded space for motion inversion. Specifically, we optimize different latent code for each frame.
> The inversed motion codes can be used for retargeting a different person. We first use the motion latent codes obtained in Sec. 4.4 to animate the static images. We also use the motion codes of the inversed videos in Sec 4.4 to drive another person’s face. The generated results can be found on our project page (the video grid with caption “Motion Retargeting”). The results show that the inversed codes are still within the distribution and they can be used for retargeting a different appearance code.

---

> > ### Comment · Reviewer_7Zbn · 2022-11-26
> > **Thanks for the response**
> >
> > I like the long video synthesis and retargeting results! While not perfect, these are impressive results that should inspire further work in this area.

---

> > > ### Author Response · Authors · 2022-11-27
> > > **Response to reviewer 7Zbn**
> > >
> > > Thanks for the feedback, we are glad that our results can address your questions.

---

> ### Author Response · Authors · 2022-11-16
> **Response to reviewer 7Zbn (1/2)**
>
> We thank the reviewer for the positive feedback and recognition that 1) our work is the first 3D GAN for portrait video synthesis; 2) our results have impressive geometry, appearance and motion; 3) our experiments are thorough. We respond to each of your comments one-by-one in what follows.
>
> > **1.1 Since the motion component only controls the first few generator layers, fine-grained motion would be difficult to model.**
>
> As we analyzed in Appendix A.2, there exists a trade-off between the video (motion) diversity and the appearance consistency. We empirically found that manipulating only the first few generator layers produces the best results. It is possible that our model may lose some capacity for the fine-grained motion, though it is hard to define the evaluation metrics for motion granularity in portrait video generation. To verify our model’s ability, we generated additional samples. The results can be found on our project page (the video grid with caption “Portrait Video Samples”). Our model can synthesize videos with fine-grained motion. For example, video (a) and (d) contain gaze motion, video (e) has a rising eyebrow, and video  (f) contains wrinkle changes on her forehead. We agree that fine-grained motion is an important factor for high-fidelity photo-realistic video synthesis. We will explore more in our future work, e.g., adopt a deformation field to model fine-grained motions.
>
> > **1.2 In addition, any lighting effects would not be modeled with the first four latents. I suspect the method does not generate any result where the lighting on the face changes between different timesteps.**
>
> Our model is able to synthesize different lighting conditions across different videos but it is not so common that lighting on faces changes between different timesteps. There are two reasons: (1) Our model is an unconditional GAN which does not disentangle the lighting condition or model it explicitly.  (2) Existing video datasets are limited in the diversity of lighting conditions. Although the CelebV-HQ dataset contains various lighting conditions, the lighting within one video clip always remains unchanged. Our generator is trained on 16 frames. Thus, the  lighting condition changes within 16 frames is even more subtle. We agree that lighting condition is an important element in the portrait video generation, we will explore how to explicitly model it in our framework in future works. In addition, we will further collect synthetic or real datasets with lighting condition changes to improve the lighting diversity.

---

### Official Review · Reviewer_Y7rC · 2022-10-30

**Confidence:** 4
**Correctness:** 3
**Technical Novelty And Significance:** 3
**Empirical Novelty And Significance:** 3
**Recommendation:** 6

**Clarity, Quality, Novelty And Reproducibility:**

The paper is well-written, and the results are good.
I think several designs proposed in this paper are effective, and the experimental results also give detailed ablation on them.

**Strength And Weaknesses:**

Strengths:
1. This paper is well-written.
2. The quality of multi-view videos is good.
3. The motion synthesis layer is well-designed. The feature modulated by the motion code are fused with apperance feature in a residual manner, which is well-motivated and resonable.

Weakness:
1. The video discriminator only receives two frames, which harm the ability to synthesis long videos. I am wondering whether this method can enable long-term video synthesis.
2. I am confused about the details of the generation pose condition. Do you simply concatenate the camera pose with latent code like EG3D? If so, are there any 2D billboard artifacts?
3. For the discriminator pose condition, how do you inject the discriminator with two poses? If the [vi, vj] are embedded into a single vector, maybe the discriminator cannot differentiate the alignment between pose pairs and image pairs.

**Summary Of The Paper:**

This paper aims to synthesize multi-view consistent portrait videos from 2D observations.
The framework is built upon recent state-of-the-art 3D-aware image GAN and aids extra-temporal information to it.
Several designs are introduced for both temporal smoothness and multi-view consistency.
The demo videos show high-quality results.

**Summary Of The Review:**

Shown in the above.

---

> ### Author Response · Authors · 2022-11-16
> **Response to reviewer Y7rC**
>
> We thank the reviewer for the positive feedback and recognition that 1) our design is well-motivated and effective; 2) our results are of high-quality and multi-view consistency is good. We respond to each of your comments one-by-one in what follows.
>
> > **1. The video discriminator only receives two frames, which harm the ability to synthesize long videos. I am wondering whether this method can enable long-term video synthesis.**
>
> We agree that the ability to synthesize long videos is important and there are some recent works [1] that focus on long video synthesis. Although our model was originally trained on short video clips (16 frames), our framework is flexible and can learn on long video clips without any network modification. To study the performance of long video synthesis, we train our model on 48-frame (3x longer) video clips (VoxCeleb). The quantitative results are shown in the table below. To study its performance, we also train a baseline model 3DVidGen on 48 frames and make comparisons between ours and 3DVidGen. As shown in the table, our model still achieves better results for long video synthesis in terms of all the evaluation metrics. We visualize the qualitative results on our [project page](https://iclr2023-pv3d.github.io/) (video grid with caption "Long video synthesis"). The visualisation shows that our generator can still synthesize reasonable long videos with good temporal consistency and diverse content.
>
> |                 | FVD&#8595;  | ID&#8593;  | CD&#8595;   | WE&#8595;   |
> |------------|-------|------|------|-------|
> | 3DVidGen   | 247.5 | 0.77 | 3.06 | 40.97 |
> | PV3D(Ours) | 152.8 | 0.80 | 1.77 | 11.44 |
> ||||||
>
> Kindly note that the strategy of using only 2 frames for discriminator training has been used in recent works [1, 2] as well. We follow them to use this strategy because of its affordable computational cost. We agree that using more frames could be helpful. We will leave this for our future work.
>
> > **2. I am confused about the details of the generation pose condition. Do you simply concatenate the camera pose with latent code like EG3D? If so, are there any 2D billboard artifacts?**
>
> Yes, the camera pose is first mapped to 512-dimension and concatenated with the latent code as done in EG3D. The concatenated code is passed to the mapping network and embedded to intermediate code space. In our video generator, we seldom observed billboard artifacts. Such artifacts are mainly caused by the local minima, which means that the generator learns to synthesize reasonable results only for the input camera viewpoints. EG3D proposed a pose swapping strategy to alleviate this issue. In our work, we also apply this trick during training, which largely alleviates the billboard artifacts.
>
> > **3. For the discriminator pose condition, how do you inject the discriminator with two poses? If the $[v_i, v_j]$ are embedded into a single vector, maybe the discriminator cannot differentiate the alignment between pose pairs and image pairs.**
>
> The poses are concatenated and embedded into a single vector. When we sample two training frames from a video clip, we keep their relative order, which means that the 1st sampled frame appears earlier than the 2nd sampled frame in the original video. So, the order of the poses $[v_i, v_j]$ is aligned with the order of the generated frames $[I_i, I_j]$ before being passed to the discriminators. To study the alignment ability of our video discriminator, we further conduct two experiments:
>
> - We sample 3000 camera pose sequences and input both the original and flipped sequences into our video discriminator to get their embeddings. Then we compute the cosine similarities between the embeddings and show the statistics (refer to Figure 8(a) in our Appendix). It shows that most of the camera pose sequences with reversed orders will be mapped into the embeddings that are apart in the embedding space, which means that our video discriminator is sensitive to the camera pose orders.
>
> - We further input the videos generated by the sampled camera poses together with both original and flipped camera pose sequences into the video discriminator. We compute the L1 distances between the real/fake scores and visualize them in Figure 8(b) in our Appendix. It illustrates that our video discriminator predicts different real/fake scores for most of the video frame pairs if the camera pose sequences are in reverse order, which means that our discriminator is order-aware and can align the camera pose sequence with the video frame pair .
>
> Please refer to our Appendix Sec. A.5 for more details.
>
> **References**
>
> [1] I. Skorokhodov, et. al. Stylegan-v: A continuous video generator with the price, image quality and perks of stylegan2. CVPR 2022.
>
> [2] S. Yu, et al. Generating videos with dynamics-aware implicit generative adversarial networks. ICLR 2022.

---

> > ### Comment · Reviewer_Y7rC · 2022-12-09
> > **Thanks for the response**
> >
> > I have carefully read the rebuttal. The authors have addressed my concerns.
> > I will keep the original score and tend to accept this paper.
> > Thanks again.

---

> > > ### Author Response · Authors · 2022-12-09
> > > **Response to reviewer Y7rC**
> > >
> > > We are glad that the concerns have been addressed. Thanks for your valuable time.

---

### Decision · Program_Chairs · 2023-01-20

**Decision:**

Accept: poster

**Justification For Why Not Higher Score:**

The work is nice, and the proposed method produces high-quality portrait videos. We are happy to accept it. However, the design of the framework is mostly based on prior works: EG3D plus common design choices for extending 2D generators for 3D generation. I do not feel the novelty high enough to justify a spotlight paper. Moreover, the proposed work only shows results on human portrait video generation, while EG3D and other video generation methods do more than human generation.

One reviewer did give a score of 10. However, the provided review does not support such a high rating. Hence, discounted.

**Justification For Why Not Lower Score:**

3D-aware portrait video generation is a relatively new topic and has a wide range of applications. The presented method makes sense. The presented results are of high quality.

**Metareview: Summary, Strengths And Weaknesses:**

The paper proposes a 3D-aware portrait video generation method. Built on top of the 3D-aware face image generator work of EG3D, the paper extends it to portrait video generation by introducing a motion branch where a motion code and timesteps are mapped to activations to modulate the core image generation branch. The results are a high-quality portrait video generator.

The paper receives 4 reviews. All of the reviewers consider the paper above the bar. They consider the presentation clear. The results are of high quality. The meta-reviewer agrees with the assessment and would like to recommend acceptance of the paper.

**Note From Pc:**

if the above contains the word "oral" or "spotlight" please see: "oral" presentation means -> notable-top-5% and "spotlight" means -> notable-top-25%. As stated in our emails, we are disassociating presentation type from AC recommendations

**Summary Of Ac-Reviewer Meeting:**

N/A